

# Topologically ordered steady states in open quantum systems

Zijian Wang⋆, Xu-Dong Dai⋆, He-Ran Wang and Zhong Wang†

Institute for Advanced Study, Tsinghua University, Beijing, China

† wangzhongemail@tsinghua.edu.cn

## Abstract

The interplay between dissipation and correlation can lead to novel emergent phenomena in open systems. Here we investigate "steady-state topological order" defined by the robust topological degeneracy of steady states, which is a generalization of the ground-state topological degeneracy of closed systems. Specifically, we construct two representative Liouvillians using engineered dissipation, and exactly solve the steady states with topological degeneracy. We find that while the steady-state topological degeneracy is fragile under noise in two dimensions, it is stable in three dimensions, where a genuine many-body phase with topological degeneracy is realized. We identify universal features of steady-state topological physics such as the deconfined emergent gauge field and slow relaxation dynamics of topological defects. The transition from a topologically ordered phase to a trivial phase is also investigated via numerical simulation. Our work highlights the essential difference between ground-state topological order in closed systems and steady-state topological order in open systems.

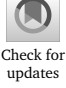

## Contents

⋆ These authors contributed equally to the development of this work.

# 1   Introduction

Topological order is one of the most fascinating quantum phases beyond the Landau paradigm [1–5]. A crucial feature that characterizes the topologically ordered phase is the topological degeneracy [2], which is robust against any local perturbation, thus providing potential use for fault-tolerant quantum computation [6]. One interesting question is how to generalize the notion of topological order when the system is coupled to an external bath. There are at least two motivations for this question. First, all physical systems are inevitably coupled to the surrounding environment. It is crucial to understand whether the robustness of topological order is guaranteed when the effect of the dissipation/noise is taken into account [7,8]. Second, new phenomena and new physics can arise from the interplay between topology and dissipation/noise in open quantum systems [9–29], which may deepen our understanding of topological physics.

The simplest case is to couple the system to a thermal bath at a finite temperature. In this case, the system will finally relax to thermal equilibrium $\rho = \frac{1}{Z}e^{-\beta H}$. Topological order in finite temperature has been diagnosed from various perspectives [8,30–35]. In this paper we aim to consider systems coupled to more general Markovian environments, described by the Lindblad equation: $\frac{d\rho}{dt} = \mathcal{L}[\rho] = i[\rho, H] + \sum_\alpha (L_\alpha \rho L_\alpha^\dagger - \frac{1}{2}\{L_\alpha^\dagger L_\alpha, \rho\})$, where $\mathcal{L}$ is called the Liouvillian superoperator, $H$ is the Hamiltonian, and $L_\alpha$ is the jump operator for channel $\alpha$. Systems under such dynamics would finally relax to a steady state, whose properties are usually of primary interest in the study of Lindblad systems. Specifically, the emergence of non-trivial many-body physics in steady states such as quantum phases and phase transitions, has been extensively explored in recent years [36–43]. The goal of this paper is to explore the possibility of non-trivial topologically ordered phase in these systems.

To this end, we propose two models with engineered dissipation, such that the degeneracy of steady states depends on the topology of the underlying lattice. As a generalization of ground-state topological order, we adopt the steady-state topological degeneracy as the defining characteristic of "steady-state topological order" (SSTO). We show that in three dimensions (3d) the topological degeneracy is stable against local perturbations to the Liouvillian, and thus a robust non-equilibrium phase with SSTO is realized. Furthermore, we diagnose more universal features of SSTO, such as the deconfined gauge field and the algebraically divergent (with respect to system size) relaxation time.

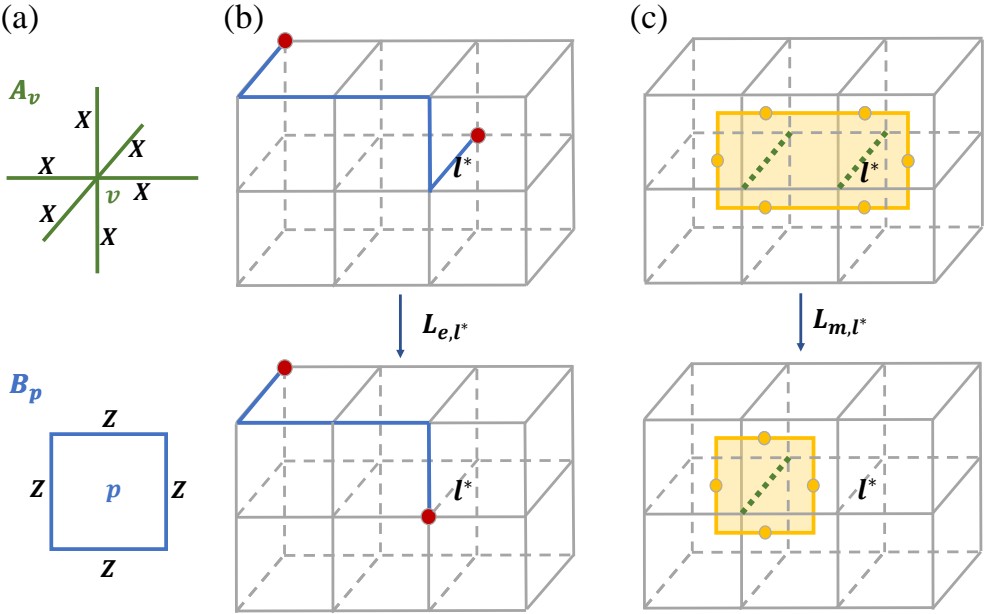

Figure 1: The effects of dissipators $L_{e,l}, L_{m,l}$. (a) $A_v$ and $B_p$. (b) links with $\sigma_l^x = -1$ are colored blue; $e$ defects living on the end of the blue string are represented by red dots. Applying $L_{e,l^*}$ on link $l^*$ in this case moves one $e$ defect to its neighboring site. (c) links with $\sigma_l^z = -1$ are colored green; $m$ defects living on plaquettes are represented by orange dots, and they form loops (orange thick lines) on the dual lattice. Applying $L_{m,l^*}$ on link $l^*$ in this case would shrink the loop defect.

## 2 Models with topological degeneracy

To realize the topological degeneracy of steady states, we construct two exactly solvable, purely dissipative ($H = 0$) lattice models defined on a $d$-torus, with the following jump operators:

$$\text{Model 1:} \quad L_{m,l} = \sigma_l^x P(\sum_{p|l\in\partial p} B_p), L_{z,l} = \sqrt{\kappa}\sigma_l^z, L_v = \sqrt{\lambda}A_v,$$

$$\text{Model 2:} \quad L_{m,l} = \sigma_l^x P(\sum_{p|l\in\partial p} B_p), L_{e,l} = \sigma_l^z P\left(\sum_{v\in\partial l} A_v\right),$$

(1)

where on each link $l$ lives a spin-$\frac{1}{2}$ degree of freedom $\sigma_l = \pm 1$. Here $A_v \equiv \prod_{l|v\in\partial l} \sigma_l^x$, $B_p \equiv \prod_{l\in\partial p} \sigma_l^z$, and $P(x)$ is a (generalized) projection operator satisfying $P(x \leq 0) > 0$ and $P(x > 0) = 0$. Clearly, the two models are inspired by the toric code model $H_{TC} = -\sum_v A_v - \sum_p B_p$, whose ground states are $2^d$-fold degenerate on a $d$-torus [6]. The $2^d$ ground states are locally indistinguishable but lie in different topological sectors. Hereafter we will focus on case $d = 3$, where the ground states can be viewed as an equal weight superposition of closed-membrane configurations,[1] and different topological sectors differ by the parity of non-contractible membranes created by $V_{ij\in\{xy,yz,xz\}} = \prod_{l\in\xi_{ij}} \sigma_l^x$, where $\xi_{ij}$ is the $ij$-plane on the dual lattice [31,44]. Note that a model similar to Model 2 was proposed to realize passive quantum error correction in [14].

---

[1]Membranes refer to the surfaces on the dual lattice formed by dual plaquettes of links with $\sigma_l^z = -1$, and $A_v$ can flip all 6 faces of a cube on the dual lattice, which is the smallest closed membrane.

We now illustrate the effect of $L_{e,l}$ and $L_{m,l}$. The lower indices $e, m$ refer to the two types of topological defects in toric code models, corresponding to vertices with $A_v = -1$ and plaquettes with $B_p = -1$ respectively. In 3d, the former are point-like, while the latter form loops (boundaries of open membranes) on the dual lattice. The effect of $L_{e,l}$ is to move one $e$ particle to the adjacent vertex or annihilate them in pairs if both ends are occupied by $e$ particles. The role of $L_{m,l}$ is more transparent when viewed on the dual lattice: they can deform/expulse the $m$ loops in a way that the loop length is always non-increasing.[2] As we will discuss later, this non-increasing feature turns out to be crucial for the stability of topological degeneracy. With this understanding, we can solve the steady states of Model 1 with the following reasoning: the $L_{m,l}$ terms lead to a state free of $m$ defects –the closed membrane states; the dephasing term $L_{z,l}$, would damp out all off-diagonal elements and lead to a diagonal $\rho$; finally the $L_v$ term would create an equal-weight mixture of closed membrane states:

$$\rho_{\text{ss}}^{\{\mu_i\}} = V_{xy}^{\mu_1} V_{xz}^{\mu_2} V_{yz}^{\mu_3} \sum_{\{v\}} \left( \prod_v A_v | \Uparrow \rangle \langle \Uparrow | \prod_v A_v \right) V_{xy}^{\mu_1} V_{xz}^{\mu_2} V_{yz}^{\mu_3}, \qquad (2)$$

where $| \Uparrow \rangle \equiv \bigotimes_l | \uparrow \rangle_l$. One can check that they indeed satisfy $\mathcal{L}[\rho_{\text{ss}}] = 0$. These states are locally indistinguishable but belong to 8 distinct topological sectors labeled by $\mu_1, \mu_2, \mu_3$ with $\mu_i = 0, 1$. On a topologically trivial manifold, such as a 3-sphere, there would be no degeneracy. Hence, the 8-fold degeneracy is indeed a faithful counterpart of topological degeneracy in dissipative systems.

The 8 steady states in the above construction are all diagonal mixed states of different closed-membrane configurations, signaling an absence of quantum coherence. We can instead design models with steady-state coherences [45], by expulsing $e$ defects as well as $m$ defects, which motivates the two jump operators in Model 2. In this case, the ground states of $H_{TC}$, i.e., $|\psi_{\{\mu_i\}}\rangle = V_{xy}^{\mu_1} V_{xz}^{\mu_2} V_{yz}^{\mu_3} \prod_v \frac{1+A_v}{2} | \Uparrow \rangle$, are dark states of both $L_{m,l}$ and $L_{e,l}$. From them we can build the steady states ($\{\mu_i\} = \{\mu_i'\}$) and steady-state coherences ($\{\mu_i\} \neq \{\mu_i'\}$):

$$\rho_{\text{ss}}^{\{\mu_i\}, \{\mu_i'\}} = |\psi_{\{\mu_i\}}\rangle \langle \psi_{\{\mu_i'\}}|. \qquad (3)$$

They span a 64-dim subspace. The degeneracy is squared due to coherences between different topological sectors.

The topological degeneracy discussed above is contributed from all defect-free states. A more careful analysis would reveal a large number of additional steady states with non-contractible loop defects. However, as we analyze in detail in D, this large degeneracy is merely accidental, and is immediately lifted once the Liouvillian is slightly tuned away from the given form. Therefore, we will neglect these states in the following discussion.

## 3 Robustness of topological degeneracy

Now an important issue is whether the topological degeneracy of steady states is robust. Only if it is robust can we identify a non-trivial topological phase. For instance, the 2d versions of both models also have topological degeneracy, with different steady states distinguished by the parity of non-contractible loops winding around the two cycles of the torus. However, intriguingly, we find that the topological degeneracy is fragile in 2d, while it is robust under weak local perturbations in higher dimensions, in accordance with previous studies on the thermal stability (fragility) of topological order [7, 8, 30, 31, 46]. This is in sharp contrast to

---

[2]Note that $L_{e,l}$ flips $\sigma_l^x$ if there are $e$ particles on the end of a link, while $L_{m,l}$ flips the dual plaquette iff the $m$ defects pass through at least two edges of it.

the ground-state topological order in closed systems where topological degeneracy is usually immune to local perturbations.

To make a better comparison with the ground state topological order, we first vectorize the density matrices in a double Hilbert space, $\rho = \sum_{mn} \rho_{mn} |m\rangle\langle n| \rightarrow |\rho\rangle\rangle = \sum_{mn} \rho_{mn} |m\rangle \otimes |n\rangle$. Correspondingly, the Liouvillian superoperator is mapped to an operator:

$$\mathcal{L} = \sum_{\alpha} L_{\alpha} \otimes L_{\alpha}^* - \frac{1}{2} L_{\alpha}^{\dagger} L_{\alpha} \otimes I - \frac{1}{2} I \otimes L_{\alpha}^T L_{\alpha}^*. \tag{4}$$

Denote the exactly solvable Liouvillians defined in Eq. 1 as $\mathcal{L}_0$. The degenerate steady states $|\text{ss}_{\alpha}^{0,R}\rangle\rangle$ are right eigenstates of $\mathcal{L}_0$ with zero eigenvalues. For later convenience, we also define the left steady states $|\text{ss}_{\alpha}^{0,L}\rangle\rangle$ as left eigenstates of $\mathcal{L}_0$, $\mathcal{L}_0^{\dagger} |\text{ss}_{\mu}^{0,L}\rangle\rangle \equiv 0$. Now we add some weak local perturbation $\mathcal{L}_0 \rightarrow \mathcal{L}_0 + \delta\mathcal{L}$. Then we can perform the standard degenerate perturbation theory to get the effective Liouvillian in the steady state subspace $S_0$ of $\mathcal{L}_0$.

$$\mathcal{L}_{\text{eff}} = P \sum_{n\in\mathbb{N}} \delta\mathcal{L} (Q \frac{1}{\lambda I - Q\mathcal{L}_0 Q} Q \delta\mathcal{L})^n P, \tag{5}$$

where $P \equiv \sum_{\mu} |\text{ss}_{\mu}^{0,R}\rangle\rangle\langle\langle\text{ss}_{\mu}^{0,L}| \equiv I - Q$ is the projection onto $S_0$. $\lambda$ is to be self-consistently determined by the eigen equation $\mathcal{L}_{\text{eff}} |\lambda\rangle\rangle = \lambda |\lambda\rangle\rangle$.

First, recall the reason for the stability of ground state topological degeneracy in closed systems: (a) The degenerate ground states lie in different topological classes, non-zero off-diagonal elements of the effective Hamiltonian only arise at extremely high order perturbation; (b) the degenerate ground states are locally indistinguishable, so the diagonal terms are all identical. Together, they lead to the conclusion that the energy-splitting under local perturbation must be exponentially small. This argument fails here because the left and right steady states are different, due to the non-Hermiticity of $\mathcal{L}^0$. Although the degenerate (right) steady states $|\text{ss}_{\mu}^{0,R}\rangle\rangle$, $|\text{ss}_{\nu}^{0,R}\rangle\rangle$ ($\mu \neq \nu$) lie in different topological classes, that does not guarantee the vanishing of $\langle\langle\text{ss}_{\mu}^{0,L}|O|\text{ss}_{\nu}^{0,R}\rangle\rangle$ for local operators $O$. This subtlety leads to the fragility of topological degeneracy in two dimensions. In B we show that the degeneracy is immediately lifted at first order. That is the reason why we focus on three dimensions, which is the lowest dimension where robust topological degeneracy is possible.

From the above analysis, it becomes clear that to realize robust topological degeneracy, configurations contained in $|\text{ss}_{\mu}^{0,R}\rangle\rangle$ and $|\text{ss}_{\nu}^{0,L}\rangle\rangle$ ($\mu \neq \nu$) are also required to differ in a highly non-local way. This is generally not easy to achieve, but our Model 1 in three (and higher) dimensions is designed to ensure that. Consequently, the 8-fold topological degeneracy will not be lifted to any finite-order perturbation, and weak local perturbations only causes an exponentially small level splitting between the topologically degenerate steady states. More details are provided in D.

The notable disparity between the behaviors in two dimensions and three dimensions can be understood through a more intuitive perspective. In two dimensions, both types of topological defects are point-like. Starting from a defect-free state in a given sector, defects can be created in pairs under local perturbation, and then they can move randomly under the Lindblad dynamics and may relatively wind around a nontrivial cycle along the torus, causing mixing between topological sectors. The loss of topological memory indicates the absence of topological degeneracy. However, in three dimensions, although small loop defects may also be created by perturbation, they tend to shrink under the dynamics due to $\mathcal{L}_0$. Under weak perturbations, the loop defects are therefore suppressed to small lengths and low densities. As a result, the topological memory, and consequently the topological degeneracy, remains robust.

Finally, for three-dimensional Model 2, the fate of point-like $e$ defects under perturbation is similar to the two-dimensional case. As a result, the degree of topological degeneracy would be

immediately reduced from 64 to 8 under weak perturbations, with the steady-state coherences destroyed but the remaining 8-fold topological degeneracy unharmed.[3] We explain in more detail how the degeneracy are lifted from 64 to 8 in E.

The robustness of the 8-fold topological degeneracy in both models allows us to identify the steady states as a non-equilibrium topological phase, which exhibits SSTO. In the rest of this paper, we attempt to identify more universal features as characteristics of SSTO and try to understand the nature of the phase transition to a trivial phase. Since the two models share the same universal features, we will focus the discussion on Model 1 for simplicity.

## 4 Deconfined dissipative gauge theory

One typical feature of topological order in Hamiltonian systems is the presence of emergent deconfined gauge fields [47–49]. For example, the toric code model realizes a deconfined phase of $Z_2$ gauge theory, and the transition to a trivial phase can be viewed as a deconfinement-confinement transition [50, 51]. Here we show that similar physics also exists in SSTO. First, notice that for both models, $\mathcal{L}_0$ is invariant under the gauge transformation generated by $A_v$: $L_\alpha \to A_v L_\alpha A_v$. The novelty in our dissipative models is that the gauge theory emerges dynamically. The initial states can be any configuration and are generally not gauge invariant, i.e., $\rho \neq A_v \rho A_v$. However, it turns out the gauge non-invariant modes would decay quickly under the relaxation dynamics. Hence, the long-time dynamics can be described by a pure gauge theory where states are gauge invariant. This feature enables us to investigate the transition from the topologically ordered phase to the trivial phase from the perspective of $Z_2$ gauge theory.

Now we focus on Model 1 for a more concrete discussion. To investigate the transition we consider the bit flip perturbation, $\delta\mathcal{L}[\cdot] = L_{x,l} \cdot L_{x,l}^\dagger - \frac{1}{2}\{L_{x,l}^\dagger L_{x,l}, \cdot\}$, with the jump operator $L_{x,l} = \sqrt{h}\sigma_l^x$. We note that the set of all diagonal matrices in the $\sigma^z$ basis forms an invariant subspace of $\mathcal{L}_0$. Because of the dephasing term $L_{z,l}$, the long-time dynamics is governed by the evolution in the subspace of diagonal matrices: $\rho = \sum_m p(m)|m\rangle\langle m|$. Then the long-time Lindblad dynamics is reduced to a classical Markov dynamics:

$$
\begin{aligned}
\frac{d}{dt}p(m) &= \sum_{n \neq m} \Gamma_{mn}p(n) - \Gamma_{nm}p(m), \\
\Gamma &= \sum_l (\sigma_l^x - 1)P^2(\sum_{l \in \partial p} B_p) + h(\sigma_l^x - 1).
\end{aligned}
\tag{6}
$$

The gauge invariance is now manifest: $[A_v, \Gamma] = 0$, and Gauss's law $A_v \rho A_v = \rho$ is already imposed.

We wonder if there exists a deconfinement-confinement transition in this non-equilibrium context. Therefore, we examine the expectation value of the Wilson loop operator in the steady state $\rho_{ss}$,

$$
\langle W_\gamma \rangle \equiv \text{tr}(W_\gamma \rho_{ss}), \quad W_\gamma = \prod_{l \in \gamma} \sigma_l^z,
\tag{7}
$$

where $\gamma$ is a closed loop. Since the limiting cases $h = 0$ and $h = \infty$ can be solved exactly, we calculate $\langle W_\gamma \rangle$ via perturbation expansion in small/large $h$ limit,[4] and find that $\langle W_\gamma \rangle$

---

[3]Although in 3d the steady state coherences are fragile, our construction can be straightforwardly generalized to 4d where both types of defects are loop-like, and then robust topological quantum memory and long-range entanglement can be realized [7].

[4]In the perturbative and numerical calculation below, we specify $P(x)$ to be $P^2(x < 0) = 1, P^2(x = 0) = \frac{1}{2}$ and $P^2(x > 0) = 0$, following the convention in Glauber dynamics. Nevertheless, the qualitative results should be independent of the specific choice of $P(x)$.

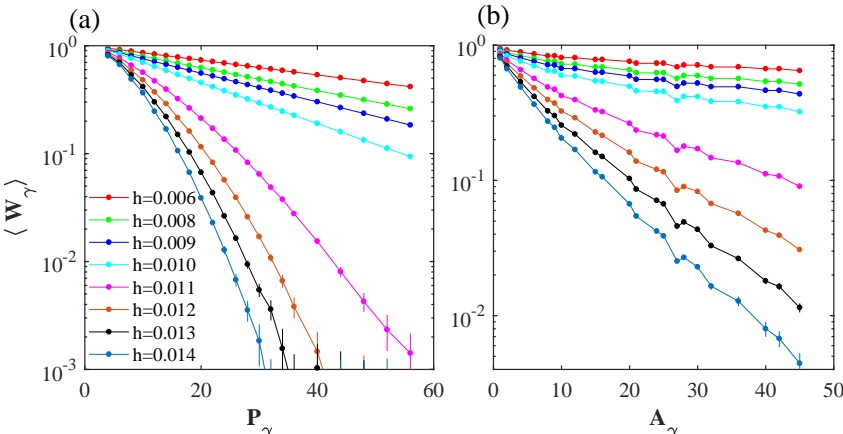

Figure 2: Calculation of the Wilson loop on a $32 \times 32 \times 32$ lattice. Semi-log plots of the expectation value of Wilson loops versus the perimeter of the Wilson loop $\gamma$ (a) and the minimal area encircled by $\gamma$ (b) are shown. Clearly, there is a deconfinement-confinement transition at $0.010 < h_c < 0.011$.

satisfies the perimeter/area law, respectively:

$$
\begin{aligned}
\langle W_\gamma \rangle &\sim \exp\left(-2hP_\gamma\right), \text{ for } h \to 0, \\
\langle W_\gamma \rangle &\sim \exp\left(-\frac{1}{2}A_\gamma \ln h\right), \text{ for } h \to \infty,
\end{aligned}
\tag{8}
$$

where $P_\gamma, A_\gamma$ is the perimeter of $\gamma$ and the area of the minimal surface enclosed by $\gamma$. Therefore, under weak perturbations, the system is in a deconfined phase, which can serve as another evidence of non-trivial topological order. We anticipate that the system will experience a deconfinement-confinement transition at a critical value of $h_c$, triggered by the proliferation of $m$ defects. This transition is expected to coincide with the breaking of the topological degeneracy. The above prediction is verified by numerical simulation of the corresponding Markov dynamics (Eq. 6). The results are shown in Fig. 2.

## 5 Slow relaxation

In the above we focus on the steady-state characteristics of the dissipative topological order. Now we discuss the long-time relaxation dynamics before the system reaches the steady state, which is also important for characterizing a non-equilibrium phase. The long-time dynamics is determined by the low-lying Liouvillian spectrum, so we can extract important features like the Liouvillian gap by investigating it. Here the Liouvillian gap is defined as the spectral gap between the topologically degenerate steady states and the rest: $\Delta = \min_{n \notin \text{steady-state subspace}} \{\text{Re}(-\lambda_n)\}$. We find that for Liouvillians with topological degeneracy, the relaxation time diverges in the thermodynamic limit. This can be understood rather intuitively: The relaxation dynamics can be viewed as the shrinking-expulsion process of loop defects, and such process must take an algebraically long time for large size of the loop defects. From the above, we expect that the Liouvillian is gapless for $h < h_c$. However, for $h > h_c$ all long-range correlations are destroyed by the proliferation of $m$ defects, and the relaxation process should be insensitive to the system size. For example, a large membrane would be immediately torn apart into pieces. In this case, the Liouvillian spectrum should be gapped. To verify this picture, we calculate the evolution of the total length of loop defects for various values of $h$, starting from random initial states. The results are shown in Fig. 3 (c). In the long time limit, the density of $m$ defects decays exponentially for $h > h_c$, signaling a finite Li-

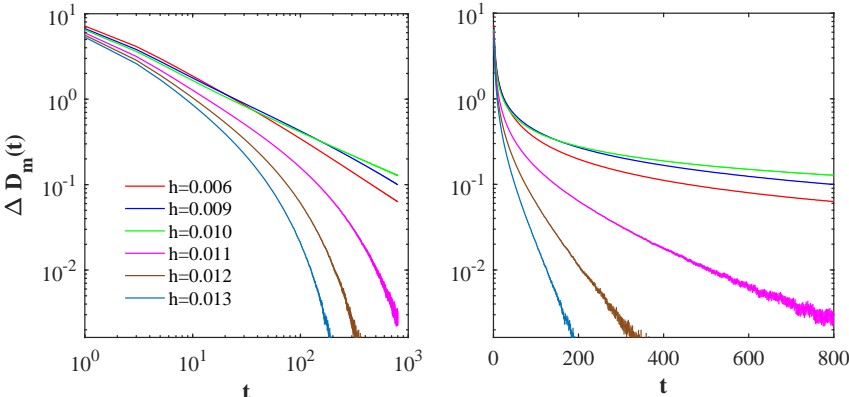

Figure 3: Evolution of the density of $m$ defects $\Delta D_m(t)$ on a $32 \times 32 \times 32$ lattice, defined as the number of plaquettes with $B_p = -1$ divided by the total number of plaquettes, with the density in steady state subtracted. Results are obtained by averaging over 96000 trajectories under random initial states. Clearly, $\Delta D_m(t)$ decays algebraically (exponentially) for $h < h_c$ ($h > h_c$).

ouvillian gap which determines the decay rate, while for $h < h_c$ the density of $m$ defects decay algebraically, which is the typical behavior for systems with a vanishing Liouvillian gap [52]. All these results are consistent with the above picture.

Recall that in closed systems topological order is defined with an energy gap separating the degenerate ground states and excitations. In our dissipative models, although the Liouvillian gap vanishes, it is algebraically small with respect to the system size, while the splitting within the topologically degenerate subspace is exponentially small. Thus, the topological degeneracy is still well-defined. Moreover, the gapless Liouvillian spectrum seems to be an inevitable consequence of topological degeneracy, and thus is a universal feature of SSTO. This reveals a sharp distinction between topological order in open systems and closed systems. We give a heuristic argument below.

In closed systems with ground state topological degeneracy, we learn that there would be topological excitations (defects) on top of the ground states [5]. They are defined as excitations that cannot be created or destroyed by local operators. In open systems, we anticipate that the system will likewise reach the topologically degenerate steady states through the expulsion of topological defects. However, since such topological defects cannot be eliminated locally, the expulsion process takes a long time for large systems, as we have already seen for loop defects in 3d in the previous discussion. As another example, we can consider point-like defects, e.g., the $e$ defects in our two models. Since they cannot disappear individually, they are only able to migrate to the vicinity of other point-like defects, where they subsequently undergo annihilation. The time this process takes also diverges when the distance between the defects is large. Due to the particular role of topological defects, the relaxation time generally diverges in the thermodynamic limit, which typically indicates the vanishing of the Liouvillian gap in the presence of translation symmetry.[5] Here the gapless mode of Liouvillian is contributed by highly non-local excitations, with extensively large $m$ defects. To further support the correspondence between topological degeneracy and gapless spectrum, we analyze the 2d version of Model 1 in G and find that the Liouvillian is gapless for $h = 0$, when there is topological degeneracy, and gapped otherwise.

---

[5]Notably, in systems without translation invariance, such as lattice models with open boundary or disorder, there are counterexamples where the relaxation time is divergent even if the Liouvillian is gapped, e.g., see [53, 54].

# 6 Discussion

Through two concrete models, we show how robust topological order can arise in open quantum systems with dissipations. Our results highlight the differences between ground-state and steady-state topological orders. Notably, whereas the ground-state topological degeneracy occurs in both two and three dimensions, its steady-state counterpart is robust only in three (and higher) dimensions. Interesting open questions include generalizing our construction to realize other types of topological phases, such as non-Abelian topological order, fracton topological order, etc, in open systems.

# Acknowledgments

We thank Xuan Zou, Zhengzhi Wu, and Hao-Xin Wang for useful discussions.

**Funding information** This work is supported by the NSFC under Grant No. 12125405, National Key R&D Program of China (No. 2023YFA1406702), and the Innovation Program for Quantum Science and Technology (No. 2021ZD0302502).

# A  Model 1 & 2 in 2d

In this appendix, we analyze the two-dimensional version of Model 1 & 2 in the main text. First, recall that in the 2d toric code model, there are two types of underlying loop structures, i.e., $Z$ loops on the original lattice and $X$ loops on the dual lattice. The ground state is an equal-weight superposition of loop configurations, and is 4-fold degenerate on a 2-torus:

$$
\begin{aligned}
|\psi_{\{\mu_i\}}\rangle &= V_x^{\mu_1} V_y^{\mu_2} \prod_v \frac{1+A_v}{2} |\Uparrow\rangle , \\
W_{x(y)} &= \prod_{l\in\tilde{\gamma}_{x(y)}} \sigma_l^x , \quad \mu_{1,2} = 0,1 .
\end{aligned}
\tag{A.1}
$$

Here we work in the $\sigma^z$ basis, and $|\Uparrow\rangle \equiv \otimes_l |\uparrow\rangle$, so different ground states are distinguished by the parity $\mu, \nu$ of non-contractible loops $\tilde{\gamma}_{x,y}$ on the dual lattice, circling around the torus. Correspondingly, $e$ and $m$ defects live on the end of these two types of strings, respectively, so both of them are point-like defects in 2d. The effect of the dissipator $L_{e,l}$ and $L_{m,l}$ are illustrated in Fig. 4. Analogous to the analysis of the 3d case in Section 2, we can exactly solve the steady states of both models on a 2-torus. For Model 1, all steady states are diagonal and the degeneracy is 4:

$$
\rho_{ss}^{\{\mu_i\}} = V_x^{\mu_1} V_y^{\mu_2} \sum_{\{\nu\}} \left( \prod_\nu A_\nu |\Uparrow\rangle\langle\Uparrow| \prod_\nu A_\nu \right) V_x^{\mu_1} V_y^{\mu_2} .
\tag{A.2}
$$

For Model 2, steady states and steady-state coherences can be constructed from $|\psi_{\mu_i}\rangle$, spanning a 16-dim steady state subspace:

$$
\rho_{ss}^{\{\mu_i\},\{\mu_i'\}} = |\psi_{\{\mu_i\}}\rangle\langle\psi_{\{\mu_i'\}}| .
\tag{A.3}
$$

Generically, the degeneracy on a genus-$g$ manifold is $4^g$ and $4^{2g}$ for the two models, respectively. We note that a similar model to Model 2 in 2d has been studied in [11].

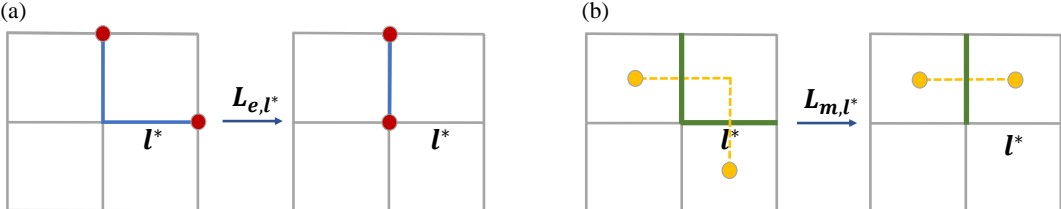

Figure 4: An illustration of the effects of dissipators $L_{e,l}, L_{m,l}$ in 2d. (a) links with $\sigma_l^x = -1$ are colored blue; $e$ defects living on the end of the blue string are represented by red dots. Applying $L_{e,l^*}$ on link $l^*$ in this case moves one $e$ defect to its neighboring site. (b) links with $\sigma_l^z = -1$ are colored green; $m$ defects living on plaquettes (vertices on the dual lattice) are represented by orange dots. Applying $L_{m,l^*}$ on link $l^*$ in this case would move one $m$ defect to its neighboring plaquette.

## B Fragility of topological degeneracy in 2d

As mentioned in Section 3, the topological degeneracy of steady states is not guaranteed to be robust, due to the complication that the left and right steady states are different in general. Although our model is designed such that the 8-fold topological degeneracy is robust in 3d, here we show that in 2d, the topological degeneracy of both models is fragile under perturbation.

We analyze Model 1 first. Parallel to the discussion in the main text, we can reduce the problem to a classical Markov dynamics, with the generator:

$$\Gamma = \Gamma^0 + \delta\Gamma = \sum_l (\sigma_l^x - 1)P\left(\sum_{p|l\in\partial p} B_p\right) + \lambda \sum_v (A_v - 1) + \sum_l h(\sigma_l^x - 1). \tag{B.1}$$

The last term $\delta\Gamma = \sum_l h(\sigma_l^x - 1)$ is treated as perturbation, and we need to find out whether the topological degeneracy persists to a finite small $h$. First, we'd like to perform the degenerate perturbation theory. To simplify the notation, we first vectorize the density matrices $\rho = p(m)|m\rangle\langle m| \rightarrow |\rho\rangle = p(m)|m\rangle$. Since we only need to consider diagonal density matrices (in the $\sigma_z$ basis) for this model, we don't need to double the Hilbert space here. In this way the master equation can be written as $\frac{d}{dt}|\rho\rangle = \Gamma|\rho\rangle$, and the right and left steady states $|ss^{R,L}\rangle$ correspond to right and left zero-energy ground states of the fictitious non-Hermitian "Hamiltonian" $-\Gamma$. Then we can obtain the effective generator in the degenerate subspace under perturbation by simply rewriting Eq. 5 in the main text.

$$\Gamma_{\text{eff}} = P\sum_{n\in\mathbb{N}} \delta\Gamma\left(Q\frac{1}{\lambda I - Q\Gamma^0 Q}Q\delta\Gamma\right)^n P, \tag{B.2}$$

where $P \equiv \sum_\mu |ss_\mu^{0,R}\rangle\langle ss_\mu^{0,L}| \equiv I - Q$ is the projection onto $S_0$. $\lambda$ is to be self-consistently determined by the eigen equation $\Gamma_{\text{eff}}|\lambda\rangle = \lambda|\lambda\rangle$. To evaluate the expansion, we need both the expression of left and right steady states of the unperturbed generator $\Gamma^0$.

$$|ss_\mu^{0,R}\rangle = V_x^{\mu_1} V_y^{\mu_2} \prod_v \frac{1+A_v}{2}|0\rangle,$$

$$|ss_{\mu'}^{0,L}\rangle = \frac{1}{\mathcal{N}} \lim_{t\to\infty} e^{\Gamma_0^\dagger t}|ss_{\mu'}^{0,R}\rangle \tag{B.3}$$

$$= \frac{1}{2} V_x^{\mu_1'} V_y^{\mu_2'} \prod_v (1+A_v)\left(|0\rangle + \sum_{\{m_2\}} \alpha_2(\{m_2\})|\{m_2\}\rangle + \sum_{\{m_2\}} \alpha_4(\{m_4\})|\{m_4\}\rangle + \cdots\right),$$

with the orthonormal condition $\langle \text{ss}^{0,L}_{\mu'}|\text{ss}^{0,R}_\mu\rangle = \delta_{\mu_1\mu'_1}\delta_{\mu_2\mu'_2}$. Here $\{m_{2k}\}$ refers to configurations with $2k$ $m$-defects. The last line follows from the fact that $\Gamma^{0\dagger}$ can move $m$ defects or create them in pairs. Thus non-zero off-diagonal elements of the effective generator $P\Gamma_{\text{eff}}P$ arise even at first order, i.e., $\langle \text{ss}^{0,L}_{\mu'}|\delta\Gamma|\text{ss}^{0,R}_\mu\rangle \neq 0$ for $\mu \neq \mu'$. In Ref. [55] we give a more quantitative analysis, where it is shown that the magnitude of the off-diagonal term is $\sim hL^2/\log L$. Hence, the 4-fold degeneracy is immediately lifted for any finite $h$, and the steady state is unique. Physically this is due to the proliferation of $m$ defects under perturbation.

Actually, under this specific form of $\delta\Gamma$, the steady state can be exactly solved:

$$|\text{ss}^R\rangle = \sum_k \left(\frac{h}{1+h}\right)^k \prod_{i=x,y} \frac{1+V_i}{2} \sum_{\{\mathbf{r}\}} |m_{2k}(\{\mathbf{r}\})\rangle. \tag{B.4}$$

Here $|m_{2k}(\{\mathbf{r}\})\rangle$ denotes states with $2k$ $m$ defects with positions $\{\mathbf{r}\} = (\mathbf{r}_1, \mathbf{r}_2, \cdots, \mathbf{r}_{2k})$, and the weight of each configuration only depends on the number of $m$ defects. One can easily check that $|\text{ss}^R\rangle$ satisfies the detailed balance condition $\Gamma_{mn}p(n) = \Gamma_{nm}p(m)$ for any pair of $n, m$. Indeed, the corresponding diagonal density matrix is just the Gibbs state $\rho_{\text{ss}} = \exp\left(-\sum_p B_p/T\right)$, with the effective temperature $T = \frac{4}{\ln\frac{1+h}{h}}$. The property of $\rho_{\text{ss}}$ has been thoroughly investigated in the study of Ising gauge theory, where it is found that in 2d the Wilson loop expectation value always satisfies an area law. This further confirms the absence of robust topological order in 2d.

Next, we're going to show that the topological degeneracy of Model 2 is also fragile. We consider the following two types of perturbation:

$$\begin{aligned} L_{x,l} &= \sqrt{h_x}\sigma^x_l, \\ L_{z,l} &= \sqrt{h_z}\sigma^z_l. \end{aligned} \tag{B.5}$$

This model is also exactly solvable based on the following reasoning: 1. The action on $m$ defects is identical to that in Model 1. 2. The action on $e$ defects is of the same form as the action on $e$ defects (identical in the case $h_x = h_z$). 3. The Liouvillian superoperator has an invariant subspace

$$A = \Big\{\rho = \sum_{mn} \rho_{mn}|m\rangle\langle n| \quad \big| \quad |m\rangle, |n\rangle \text{ contain the same } e \text{ and } m \text{ defect configuration}\Big\}.$$

We can seek steady states within this subspace. 4. Although $e, m$ defects have nontrivial mutual statistics in Hamiltonian systems, any phase factor from braiding would cancel out under the Lindblad dynamics in $A$. Therefore, the dynamics of $e/m$ particles are completely independent.

The key message gained from the above analysis is that this model can be viewed as a double version of Model 1 (in terms of the dynamics of defects), so we expect the topological degeneracy to be completely lifted when both $h_x$ and $h_z$ are nonzero. We can directly get the exact steady state by generalizing the results of Model 1: $\rho_{\text{ss}} = \exp\left(-\sum_p B_p/T_m - \sum_v A_v/T_e\right)$ with $T_{m(e)} = \frac{4}{\ln\frac{1+h_{x(z)}}{h_{x(z)}}}$. This is just the thermal state of the toric code model. It is known that it has no topological order under finite temperature in 2d, consistent with our expectations.

Neither of the models we construct exhibits robust topological degeneracy in 2d. We give a more intuitive argument below. First, note that both types of topological defects in 2d are point-like, and the topologically degenerate steady states in the ideal models are reached via the expulsion of these defects. However, once some weak noise/perturbation is present, defects can be created in pairs, and there is nothing preventing them from relatively winding around a large circle of the torus. Hence, even a single pair of defects can erase the memory of the initial topological sector. We believe this picture is generically true for 2d Lindblad systems.

(a) 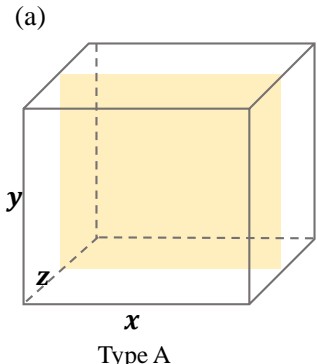 (b) 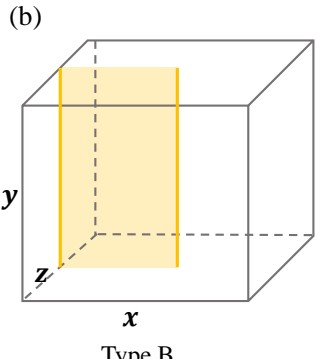

Type A                    Type B

Figure 5: (a) A typical configuration of type A states, where there are no defects and all membranes are closed. Here we show an example with a large non-contractible membrane in the $xy$ plane. (b) A typical configuration of type B states, where $m$ defects (orange solid line) form flat non-contractible loops.

## C  Accidental degeneracy in 3d

The topological degeneracy of both models in 3d is from the multiple defect-free states on 3-torus. From the view on the dual lattice, this means only closed-membrane configuration remains. In this appendix we explain that for the particular form of the two models, there are a large number additional steady states.

In fact, $L_{m,l}$ also allows some states with $m$ defects as its dark states, with the requirement that the $m$ defects form non-contractible flat loops. That follows from the strong constraint that the length of loop defects is non-increasing, and for these flat loop defects, any local move of the defect would increase the defect length. Thus there is an exponentially large number of additional steady states with the remaining $m$ defects forming non-contractible flat loops. For later convenience, we will call these steady states type B states and the defect-free steady states type A. See Fig. 5 for an illustration of these two types of steady states. From the above analysis, we can see that the existence of type B states in the steady-state subspace heavily relies on the strong constraint of the dynamics and should be merely metastable. Once the constraint is released under perturbation, we expect such metastable states would eventually evolve into the type A steady states and the large degeneracy will be lifted, as will be verified in the next appendix.

## D  Degenerate perturbation theory of Model 1 in 3d

In this appendix we apply the degenerate perturbation analysis to Model 1 in 3d. Firstly, we aim to understand the robustness of topological degeneracy in 3d, so we focus we first focus on the effective Liouvillian $P_A \mathcal{L}_{\mathrm{eff}} P_A$ in the subspace spanned by type A steady states (subspace with topological degeneracy), where $P_A$ is the projection to this subspace. The subtlety of applying the degenerate perturbation theory to Liouvillians arises from the disparity of left and right eigenstates, which causes fragility of topological degeneracy in 2d. As pointed out in Section 3, to realize robust topological degeneracy, we require,

$$\langle\!\langle \mathrm{ss}_\mu^{0,L} | O | \mathrm{ss}_\nu^{0,R} \rangle\!\rangle = 0, \quad \forall \text{ local operators } O \text{ acting on the double Hilbert space.} \tag{D.1}$$

We show below this is indeed the case for Model 1.

Notice that $|ss_\mu^{0,L}\rangle\!\rangle = \lim_{t\to\infty} e^{\mathcal{L}_0^\dagger t}|ss_\mu^{0,R}\rangle$. That is, the left steady states can be obtained by evolving the right steady states under the adjoint Liouvillian $\mathcal{L}_0^\dagger$. Contrary to $\mathcal{L}_0$, $\mathcal{L}_0^\dagger$ deform/create the loop defects in a way that the defect length never decreases. Starting from the topological sector $\mu$, loop defects can be created and grow larger, but this process cannot lead to or even get close to a defect-free state in another topological sector due to the constraint. In other words, one cannot grow a large non-contractible membrane without shrinking the boundary of the membrane. Therefore, the condition Eq. D.1 is satisfied. Consequently, all off-diagonal terms in $P_A\mathcal{L}_{\text{eff}}P_A$ vanish in any finite order perturbation and are thus exponentially small. Also, the diagonal terms in $P_A\mathcal{L}_{\text{eff}}P_A$ are identical, which follows from the fact that the 8 type A states are locally indistinguishable. These facts tell us that the 8-fold topological degeneracy of type A states cannot be lifted by any finite order perturbation, with exponentially small splitting under weak local perturbations.

Next, to understand the fate of type B states, we give a more elaborate analysis of the full effective Liouvillian:

$$\mathcal{L}_{\text{eff}} = \begin{pmatrix} P_A\mathcal{L}_{\text{eff}}P_A & P_A\mathcal{L}_{\text{eff}}P_B \\ P_B\mathcal{L}_{\text{eff}}P_A & P_B\mathcal{L}_{\text{eff}}P_B \end{pmatrix}. \tag{D.2}$$

Similar to the above discussion, we find all elements of $P_B\Gamma_{\text{eff}}P_A$ are also exponentially small, which further confirm the stability of topological degeneracy. On the contrary, for the remaining two blocks, nonzero elements arise even at the first order. This result reflects that a type B configuration is easy to turn into a type A configuration or other type B configurations under perturbation, while a type A configuration will stay in the same topological sector under small perturbation. Therefore, the additional degeneracy contributed by type B states is easily lifted, in agreement with our intuition that type B configurations are metastable. Consequently, only the 8-fold topological degeneracy is robust. In F we further study the lifetime of type B states through numerical simulations.

We must point out that $\mathcal{L}_{\text{eff}}$ cannot really determine the long-time dynamics, because the steady state subspace of $\Gamma^0$ is not well separated from the rest states due to the absence of a Liouvillian gap. Then there may be concerns that the whole argument break down. Since in Hamiltonian systems, the perturbation expansion of the effective Hamiltonian is normally controlled by $h/\Delta$ with $\Delta$ the energy gap, it obviously fails when $\Delta \to 0$. However, as we will see in G, the low-lying Liouvillian spectrum is contributed by highly non-local excitations, with extensively large $m$ defects, which only arise in extremely high-order perturbation terms. Thus the perturbation expansion is well-controlled at any finite order. We believe the above argument does give qualitatively correct results. The above argument reveals another key difference between steady-state topological order (SSTO) and ground-state topological order: the robust topological degeneracy is not protected by a Liouvillian gap but by the extensiveness of gapless modes.

# E   Fragility of steady state coherences in 3d

The robust topological degeneracy of Model 1 in 3d indicates a classical topological memory: starting from one topological sector, the system would stay in the same sector for an exponentially long time, which can be viewed as an example of classical bits. However, starting from some coherent superposition of states in different topological sectors, the coherence would be washed out and only the relative weight is preserved. On the contrary, in Model 2 phase coherences can be preserved, since all pure states within the ground state subspace of $H_{TC}$ are steady states of Model 2. Unfortunately, the steady-state coherences would be destroyed under generic perturbations, as is pointed out in Section 3.

To illustrate this point, first, we note that there are two underlying structures in the 3d toric code model: the ground state can be viewed as either a condensate of closed membranes on the dual lattice or of closed loops on the original lattice. $m, e$ are the corresponding defects of the two structures, respectively. While in our previous discussion, we choose a representation where the former structure is manifest, it turns out to be helpful to switch to the alternative representation to understand the effect of perturbation. That is, we can write the ground states of $H_{TC}$ as:

$$|\psi_{\{\tilde{\mu}_i\}}\rangle = W_x^{\tilde{\mu}_1} W_y^{\tilde{\mu}_2} W_z^{\tilde{\mu}_3} \prod_p \frac{1+B_p}{2} |\Rightarrow\rangle, \qquad (E.1)$$

where $|\Rightarrow\rangle \equiv \bigotimes_l |\rightarrow\rangle_l$ and $W$ is the non-contractible loop operator introduced in Eq. A.1. In this representation different topological sectors are distinguished by the parity of the non-contractible loops created by $W_{x,y,z}$ and any superposition of the above states are steady states of Model 2. In other words, in the double Hilbert space, the type A steady state subspace of $\mathcal{L}_0$ is spanned by $|ss_{\tilde{\mu},\tilde{\nu}}^{0,R}\rangle\rangle = |\psi_{\{\tilde{\mu}_i\}}\rangle \otimes |\psi_{\{\tilde{\nu}_i\}}\rangle$.

Now we add the perturbation terms in Eq. B.5 and see whether the topological degeneracy is robust or not. $L_{x,l}$ is the noise term causing fluctuation of $m$ defects while $L_{z,l}$ causes fluctuation of $e$ defects. As is illustrated in Model 1, the former doesn't lead to the proliferation of $m$ defects unless $h_x$ exceeds some finite threshold. However, the dynamics of $e$ defects is similar to that in 2d, so using the same argument in B, we can conclude that under any finite $h_z$, the creation and random motion of $e$ defects would cause mixing between sectors with different $\{\tilde{\mu}\}$. That means $\langle\langle ss_{\tilde{\mu}',\tilde{\nu}'}^{0,L}|\delta\mathcal{L}|ss_{\tilde{\mu},\tilde{\nu}}^{0,R}\rangle\rangle$ is in general nonzero. Then the 64-fold degeneracy would be lifted even at first-order perturbation.

In the case $h_x = 0$, the model is exactly solvable with 8 steady states:

$$\rho_{ss}^{\{\mu_i\}} = V_{xy}^{\mu_1} V_{xz}^{\mu_2} V_{yz}^{\mu_3} \left\{ \sum_k \left(\frac{h_z}{1+h_z}\right)^k \prod_{i=x,y,z} \frac{1+W_i}{2} \sum_{\{\mathbf{r}\}} |e_{2k}(\{\mathbf{r}\})\rangle\langle e_{2k}(\{\mathbf{r}\})| \prod_{i=x,y,z} \frac{1+W_i}{2} \right\} V_{xy}^{\mu_1} V_{xz}^{\mu_2} V_{yz}^{\mu_3}. \qquad (E.2)$$

Here $|e_{2k}(\{\mathbf{r}\})\rangle$ denotes states with $2k$ $e$ defects with positions $\{\mathbf{r}\} = (\mathbf{r}_1, \mathbf{r}_2, \cdots, \mathbf{r}_{2k})$. One can verify that $W_{z,y,x}\rho_{ss} = \rho_{ss}W_{z,y,x} = (-1)^{\mu_{1,2,3}}\rho_{ss}$, which means the bra and ket in the density matrix always lie in the same topological sector $\{\mu\}$ in the membrane picture. This tells us the coherence between different sectors is lost because of the creation and moving of $e$ defects. Then under any finite $h_z$, we can at most realize three classical bits that store the topological information, and all qualitative properties of steady states are identical to those of Model 1. Actually, the above steady states are exactly reduced to Eq. 2 in the main text in the limit $h_z \to \infty$. Then we can deduce from the analysis of Model 1 that the effect of small $h_x$ is to create a low density of small $m$ defects, but cannot lead to a proliferation of them, so the 8-fold topological degeneracy is preserved. Just as in Model 1, a classical topological order is realized.

From the above discussion, we learn that the dimension $d_{\text{def}}$ of the topological defects plays a crucial role in topological order in open quantum systems, only when $d_{\text{def}} \geq 1$ is the corresponding topological information robust against noise. To realize robust quantum topological order, we need to generalize Model 2 to 4d where both types of defects are loop-like. Then robust topologically degenerate steady states and steady-state coherences can be realized, with the dimension of steady state subspace $= (2^4)^2 = 256$.

# F Lifetime of type B states

In D we argue that the type B states are metastable and would finally evolve into the type A states under generic perturbations. Clearly, this requires the non-contractible loop defects

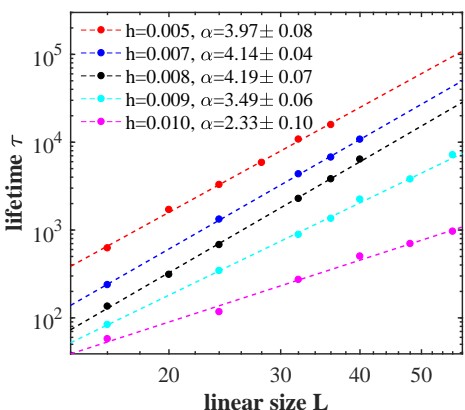

Figure 6: The lifetime of the type B configuration scales as a power law with system size. The results are obtained by averaging over 1000 trajectories for each parameter. Data obtained from numerical simulation is represented by "*" and dashed lines are obtained from linear fitting of the log-log plot. We use $\alpha$ to denote the fitted slope, so lifetime $\tau \propto L^{\alpha}$.

to merge into contractible ones, and such process takes a very long time when the distance between non-contractible loops is large. Then it is crucial to check that these states do not have an exponentially long lifetime, because only then are they well-separated from the 8-dimensional topologically degenerate subspace. Otherwise the topological degeneracy would be ill-defined.

To investigate the lifetime of the metastable states, we numerically simulate the dynamics for several parameters in the topologically ordered phase, with a membrane on the half $xy$ plane in the initial state (See Fig. 5(a)). As depicted in Fig. 6 it turns out the lifetime (practically defined as the average time when the number of non-contractible loops drops to zero) diverges algebraically with the system size: $\tau_{\text{life time}} \propto L^{\alpha}$, with $\alpha \approx 4$ for $h$ well below the critical point. Moreover, even for contractible loop defects, the shrinking-expulsion process also takes an algebraically long time when the size of the defect is large. The slow relaxation of large defects is expected to contribute to the low-lying Liouvillian spectrum above the 8-fold degenerate steady states, with an algebraically small Liouvillian gap.

## G    More on the connection between slow relaxation dynamics and topological degeneracy

In Section 5 we point out that when the steady states are topologically degenerate, the relaxation time to reach the steady state subspace would diverge in the thermodynamic limit. We regard the slow relaxation dynamics as a universal feature of SSTO, and we also give a heuristic argument based on the nature of topological defects. In this appendix, we provide more evidence to substantiate our assertion, by a more quantitative analysis of Model 1 in both 2d and 3d.

### G.1    d=2

First, we discuss the relaxation dynamics of Model 1 in two dimensions, starting with the Markov generator in Eq. B.1. Although there is no robust topological degeneracy in 2d, we can still gain some insight by comparing the case $h = 0$ (with topological degeneracy) and

$h \neq 0$ (with no topological degeneracy). Let's study the $h = 0$ case first. We neglect the fact that $m$ particles can only come in pairs for a moment, and consider the dynamics of a single $m$ particle (alternatively, we can create two $e$ particles with infinite separation so they don't see each other). Then its dynamics is simply a random walk, whose Markov generator is nothing but a tight-binding Hamiltonian, with dispersion $\lambda_k = -2 + \cos(k_x) + \cos(k_y)$. Hence we conclude that the Liouvillian gap $\Delta \sim L^{-2}$ at large $L$. Starting from a random configuration, the $m$ defects will walk randomly and annihilate in pairs, in the end, all of them will be eliminated in the steady state, and the typical relaxation time of this process is $\tau \propto L^2$.

To see if this slow relaxation behavior is related to the topological degeneracy, we now turn on a finite $h$ and lift the topological degeneracy. What is the Liouvillian gap in this case? At a time scale much longer than $\lambda^{-1}$, configurations equivalent up to transformation generated by $A_v$ are thoroughly mixed. Then the longtime dynamics can still be reduced to the dynamics of $e$ particles. That's exactly what we do in case $h = 0$. To describe the long-time dynamics, we restrict the discussion to the subspace $A_v = 1$. This simplification is reasonable if we only concern the spectrum of $-\Gamma$ lying below $\lambda$. In this case, we get an effective non-Hermitian $Z_2$ gauge theory, as promised in the Section 4. Analogous to the well-known duality between $Z_2$ gauge theory and the transverse Ising model in 2d, this non-Hermitian gauge theory is also dual to a simpler spin model on the dual lattice with $B_p = -1 \rightarrow |\uparrow\rangle$, and $B_p = +1 \rightarrow |\downarrow\rangle$.

$$-\Gamma \rightarrow H = -\sum_{\langle ij \rangle} \left( \sigma_i^- \sigma_j^- + \frac{\sigma_i^- \sigma_j^+}{2} + \frac{\sigma_i^+ \sigma_j^-}{2} \right) + 2\sum_i n_i - h \sum_{\langle ij \rangle} (\sigma_i^x \sigma_j^x - 1), \qquad \text{(G.1)}$$

where the spin degrees of freedom are defined on the vertices of the dual lattice and $n_i = \frac{1+\sigma_i^z}{2}$. More strictly speaking, $-\Gamma$ should be mapped to $PHP$ if we consider the periodic boundary condition, where $P$ is the projector to subspace with $\prod_i (-\sigma_i^z) = 1$. That is, there can only be an even number of up spins in the dual model. In the discussion below, we will not write the projector $P$ explicitly, but all the results are projected to the even parity sector implicitly. We can directly write down the ground states of $H$ based on Eq. B.4:

$$|\rho\rangle = \bigotimes_i (|\downarrow\rangle_i + \beta |\uparrow\rangle_i), \qquad \text{(G.2)}$$

where $\beta = \sqrt{\frac{h}{h+1}}$. Like all Markov generators whose steady state satisfies detailed balance, $H$ can be mapped to a hermitian Hamiltonian through a similarity transformation.

$$H_s = SHS^{-1}, \quad S = \beta^{-\sum_i n_i/2}. \qquad \text{(G.3)}$$

Dropping the constant term, we get

$$H_s = (2h+1) \left\{ -\sum_{\langle ij \rangle} \left[ \frac{1}{2}(1+\eta)\sigma_i^x \sigma_j^x + \frac{1}{2}(1-\eta)\sigma_i^y \sigma_j^y \right] + h_z \sum_i \sigma_i^z \right\}. \qquad \text{(G.4)}$$

Here $\eta = \frac{2\sqrt{h(h+1)}}{2h+1}$ and $h_z = \frac{2}{2h+1}$. This is an anisotropic XY model with a magnetic field in $z$ direction, which has been well studied decades ago [56]. We find the parameters in our model just lie on the curve $\eta^2 + (\frac{h_z}{2})^2 = 1$. For $h \neq 0$, this model is in the gapped Ising ordered phase. Since $H_s$ has the same spectrum as $H$, there is always a finite Liouvillian gap for nonzero $h$. Therefore, the threshold for a Liouvillian gap opening and topological degeneracy breaking coincide at $h = 0$.

## G.2   d=3

Next, we turn to the more interesting case in 3d, where there is really a topologically ordered phase with topological degeneracy for $h < h_c$. Here we will show that the relaxation time also diverges algebraically in the topologically ordered phase. We have already seen in F that the meta-stable type B states have algebraically long lifetime $L^\alpha$. As pointed out in Section 5, the shrinking dynamics of large contractible loop defects can also take a very long time. Here we give the scaling between the relaxation time and the size of the loop defects in the simplest but illuminating case $h = 0$.

Recall that for $h = 0$, the steady state is an equal-weight superposition of all closed-membrane states, with no loop defect. Suppose we create a large loop defect in the initial state, then this loop would shrink and finally disappear to reach the steady state. The relaxation time of this process would obviously diverge as the initial size of the loop defects goes to infinity. To estimate the scaling relation between the relaxation time $\tau$ and the initial size of the loop defects $R_0$, we'd like to make a coarse-grained description of the dynamics, where we imagine the loop to be some smooth curve. Then how does this curve evolve and shrink? The key observation is that the shrinking of the loop defect is driven by the curvature of the loop. A "flat loop" would never shrink in our model, which is exactly why the meta-stable type B states arise on a 3-torus. Second, the shrinking of the loop is blind to the sign of the curvature $K$, so only the absolute value of it is relevant. See Fig. 7. If we assume the local shrinking rate $\sigma$ (defined as the change of the loop length per unit time) is some smooth function of $K$, then the most general form of $\sigma$ is $\sigma = \sum_{n\in N_+} a_{2n} K^{2n}$. For simplicity, assume the coarse-grained loop is a circle of radius $R$, then $K = 1/R$. We have $\sigma \approx a_2 R^{-2}$ for a large $R$. Then we can obtain the evolution of $R$ as a function of $t$:

$$
\begin{aligned}
\frac{dR(t)}{dt} &\approx -2\pi R(t)\sigma , \\
\Rightarrow R(t)\frac{dR(t)}{dt} &\approx -2\pi a_2 , \\
\Rightarrow R(t) &\approx \sqrt{R_0^2 - 2\pi a_2 t} \quad \left(t < \frac{R_0^2}{2\pi a_2}\right).
\end{aligned}
\tag{G.5}
$$

Therefore, we obtain an estimation of the relaxation time $\tau \propto R_0^2$. We expect this prediction to hold for a loop defect of a generic shape, where $R$ is replaced by the perimeter $P$ of the loop.

To confirm this prediction, we perform a numerical simulation where we put a $R_0 \times R_0$ square membrane in the initial states (everywhere else is free of defects), then let it evolve according to the classical Markov generator $\Gamma^0$. By averaging over 10000 trajectories we obtain the evolution of the expectation value of loop length, as shown in Fig. 7(b). Indeed, we find the square of the loop length decay linearly with time, and the relaxation time $\tau \propto R_0^2$ (Fig. 7(c)). Here $\tau$ is obtained by taking an average of the time that the loop length drops to zero for different trajectories.

Using these results and the fact that $\Gamma^0$ ($\Gamma^{0\dagger}$) keeps the loop length non-increasing (non-decreasing), we can formally write the low-lying left and right eigenmodes of $\Gamma^0$.

$$
\begin{aligned}
|n^R\rangle &= \sum_{\substack{\text{configuration}\{m\} \\ \text{with loop length } =4R_0}} \alpha_{\{m\}}|\{m\}\rangle + (\text{configuration with loop length} < 4R_0) , \\
|n^L\rangle &= \sum_{\substack{\text{configuration}\{m\} \\ \text{with loop length } =4R_0}} \alpha_{\{m\}}|\{m\}\rangle + (\text{configuration with loop length} > 4R_0) , \\
\Gamma^0|n^R\rangle &= \lambda_n|n^R\rangle , \quad \Gamma^{0\dagger}|n^L\rangle = \lambda_n|n^L\rangle .
\end{aligned}
\tag{G.6}
$$

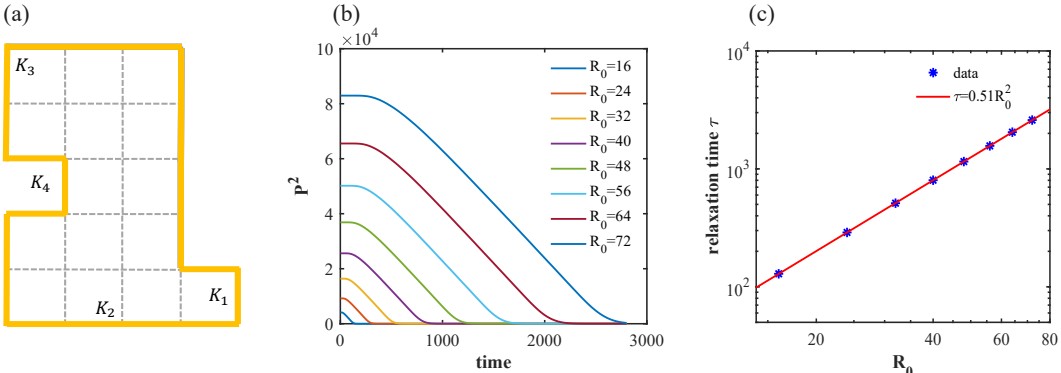

Figure 7: (a) An example of the loop defect before coarse-graining (marked in orange). The local curvature $K$ satisfies: $-K_1 = K_4 < 0 = K_2 < K_3 < K_1$; correspondingly, the local shrinking rate $\sigma$ satisfies: $0 = \sigma_2 = \sigma_3 < \sigma_1 = \sigma_4$, so $\sigma$ only depends on the absolute value of $K$. (b)-(c) Relaxation dynamics at $h = 0$ with a $R_0 \times R_0$ square membrane in the initial state. The evolution of the square of the loop perimeter $P$ is shown in (b). For sufficiently large $t$ but still $t \ll R_0^2$, we observe a wide range of time that $P^2$ drops linearly with time. The slope is nearly identical for all values of $R_0$, which leads to $\tau \propto R_0^2$. In (c) we show the scaling relation between relaxation time and $R_0$. Indeed we find $\tau \propto R_0^2$.

For large $R_0$, $\lambda_n \propto R_0^{-2}$, so low-lying eigenmodes with $\lambda_n$ close to zero are contributed by highly non-local excitation modes with extensively large $R_0$. As we have mentioned in D, such modes only contribute to extremely high-order perturbation terms in Eq. B.2.

Based on this understanding, we expect that similar things also happen in other regions of the topologically ordered phase ($0 < h < h_c$), where the loop defects don't proliferate in the steady state. The shrinking process of a large open membrane would still happen slowly (diffusively) and dominate the long-time dynamics. Indeed, we have seen in Section 5 that the deviation of total loop length decays algebraically in the entire topologically ordered phase.

# H  Perturbative calculation of the Wilson loop

In this appendix, we aim to calculate the Wilson loop

$$\langle W_\gamma \rangle = \text{tr}(W_\gamma \rho_{ss}) = \frac{\langle I | W_\gamma | ss \rangle}{\langle I | ss \rangle}, \tag{H.1}$$

where $|ss\rangle$ is the vectorization of the right steady states $\rho_{ss}$ and $|I\rangle = \otimes_l |\rightarrow\rangle$ is the vectorization of the identity operator, which is the left steady state. Here we perform the calculation via perturbation expansion: $|ss\rangle = \sum_n |ss^{(n)}\rangle$ in the limit of small and large $h$, following similar strategies as in thermal equilibrium [51].

First, we analyze the case $h \ll 1$ and treat the $\sigma_l^x$ term as a perturbation. Without loss of generality, we choose the unperturbed steady state as the one in the trivial topological sector, that is, without the non-contractible membranes:

$$|ss^{(0)}\rangle = \prod_s \frac{1 + A_s}{2} |\Uparrow\rangle. \tag{H.2}$$

Then we perform the perturbation expansion:

$$(\Gamma^0 + \delta\Gamma)\sum_n |\text{ss}^{(n)}\rangle = 0$$

$$\Rightarrow \quad 1^{st} \text{ order}: \quad \Gamma^0|\text{ss}^{(1)}\rangle + \delta\Gamma|\text{ss}^{(0)}\rangle = 0,$$
$$2^{nd} \text{ order}: \quad \Gamma^0|\text{ss}^{(2)}\rangle + \delta\Gamma|\text{ss}^{(1)}\rangle = 0, \qquad \text{(H.3)}$$
$$\cdots$$
$$n^{th} \text{ order}: \quad \Gamma^0|\text{ss}^{(n)}\rangle + \delta\Gamma|\text{ss}^{(n-1)}\rangle = 0,$$
$$\cdots$$

We use loop configuration (boundaries of open membranes) $C$ on the dual lattice to represent the solution. The (unnormalized) first-order contribution is easy to obtain:

$$|\text{ss}^{(1)}\rangle = h \sum_{\text{one 4-loop}} |C\rangle. \qquad \text{(H.4)}$$

Here "$n$-loop" represents a loop defect of length $n$ and the summation is over all possible specified $n$-loop (here n=4) configuration. At second order, the result is already rather complicated, with the following form:

$$|\text{ss}^{(2)}\rangle = \alpha^{(2)} \sum_{\text{two independent 4-loops}} |C\rangle + \sum_{\text{two adjacent 4-loops}} \beta_C |C\rangle$$
$$+ \sum_{\text{one 8-loop}} \gamma_C |C\rangle + \sum_{\text{one 6-loop}} \delta_C |C\rangle + \epsilon_C \sum_{\text{one 4-loop}} |C\rangle. \qquad \text{(H.5)}$$

Here by "independent" we mean loops that won't be fused together to larger loops by applying $\Gamma^0$. Although the full expression is difficult to obtain, we can get $\alpha^{(2)}$ easily, by noting that to second order, the weight coefficient of any two independent 4-loops in $\Gamma^0|\text{ss}^{(2)}\rangle$ and $\delta\Gamma|\text{ss}^{(1)}\rangle$ should cancel, that is, $2h^2 - 2\alpha^{(2)} = 0$, so $\alpha^{(2)} = h^2$. To $n^{th}$ order, the situation is similar:

$$|\text{ss}^{(n)}\rangle = h^n \sum_{n \text{ independent 4-loops}} |C\rangle + \cdots, \qquad \text{(H.6)}$$

where "$\cdots$" contains terms with all other possible loop configurations with a total loop length not larger than $4n$. We know little about the weight coefficient of these terms, except that they are of order $O(h^n)$. Fortunately, this is sufficient for us to calculate the leading contribution to the expectation values of the Wilson loops. Denote the total number of links by $N$ and the length of $\gamma$ by $P$, and consider the limit $N, P \to \infty$. We calculate the numerator in Eq. H.1 first. For each configuration $W_\gamma = 1 \ (-1)$ if $\gamma$ crosses an even (odd) number of loop defects. Our strategy is to perform series expansion of $h$: $\langle I|W_\gamma|\text{ss}\rangle = \sum_n w_n h^n$, and at each order $n$, we only keep the leading order term of $N$ and $P$, that is, we only keep leading order terms $N^a P^b$ with $a + b = \sup(a + b) = n$. In this approximation, the $n^{th}$-order contribution to $\langle I|W_{\gamma_c}|\text{ss}\rangle$ is dominated by the first term in Eq. H.6. Then we can write down the explicit form of it:

$$\langle I|W_\gamma|\text{ss}\rangle \approx \sum_n \frac{1}{n!}(N - P - P)^n h^n = \exp[-h(N - 2P)]. \qquad \text{(H.7)}$$

The same strategy can be applied to the calculation of the denominator:

$$\langle I|\text{ss}\rangle \approx \sum_n \frac{1}{n!} N^n h^n = \exp[-hN]. \qquad \text{(H.8)}$$

In the end, we get a very simple result:

$$\langle W_\gamma \rangle = \frac{\langle I | W_\gamma | \mathrm{ss} \rangle}{\langle I | \mathrm{ss} \rangle} = \exp(-2hP). \tag{H.9}$$

That tells us that the Wilson loop satisfies a perimeter for small $h$, which is known as the criterion of a deconfined phase.

Next, we discuss the opposite limit $h \gg 1$, then the other term should be regarded as the perturbation. The unperturbed steady state is:

$$|\mathrm{ss}^{(0)}\rangle = \bigotimes_l | \rightarrow \rangle_l. \tag{H.10}$$

The expectation value of any Wilson loop operator in this state is 0. Denote the minimal area of the membrane enclosed by $\Gamma$ as $A$. Each order of perturbations can create at most two more plaquettes in $A$ with $| \leftarrow \rangle$. Then at least to order $(\frac{1}{h})^{A/2}$ would we get non-zero contributions to $\langle W_\gamma \rangle$. Therefore, the expectation value can be estimated as:

$$\langle W_\gamma \rangle \sim h^{-A/2} = e^{-\frac{1}{2} A \ln h}. \tag{H.11}$$

Hence for large $h$, the system is in a confined phase.

The qualitative behavior of the two extreme limits is indeed verified by the numerical simulation, as shown in Fig. 2 in the main text.

# I  Numerical methods

In this appendix, we briefly introduce the numerical methods we use to study Model 1. In fact, we numerically simulate the discrete version of the Markov dynamics generated by $\Gamma$, using the classical Monte Carlo method. For each step we randomly choose one link, say $l$, and flip the spin $\sigma_l$ with the probability:

$$\mathrm{Prob}(h) = \begin{cases} \frac{h}{1+h}, & \text{if the flip decreases } \sum_{l \in \partial p} B_p, \\ \frac{0.5+h}{1+h}, & \text{if the flip keeps } \sum_{l \in \partial p} B_p \text{ invariant}, \\ 1, & \text{if the flip increases } \sum_{l \in \partial p} B_p. \end{cases} \tag{I.1}$$

Here the flipping process from the $A_v$ term is neglected since we only calculate gauge-invariant observables which are not affected by such process. To calculate the steady-state expectation value of observables, we start from some random configuration and let the system evolve a sufficiently long time (defined as the number of steps divided by the system size) $t_w$ to reach the steady states, then take the time average for $t > t_w$. To calculate dynamical quantities such as time evolution or relaxation time, we duplicate many copies of the system and take averages over different trajectories.

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
