# Peer review of "Topologically ordered steady states in open quantum systems"

_SciPost Physics, doi:SciPost Phys. 17, 167 (2024)_

## Round 1 · Referee Report · Anonymous (Referee 1) · 2024-11-2

Report

This work studied solvable models of steady-state topological orders, investigated their stability under perturbations, and also discussed confinement transition as well as relaxation properties. The result is significant, complete, and solid. I strongly recommend publishing this article in SciPost Physics.

I am curious about one question that I hope the authors could address. In the discussion of confinement and deconfinement, the authors use the expectation values of Wilson loops as a diagnostic. In conventional Lorentz invariant field theories, Wilson loop vacuum expectation values are related to the interaction strength between gauge charges via a Euclidean spacetime rotation, and thus indicates confinement/deconfinement (defined as whether gauge charges have long-range interactions or not). In Lindbladian dynamics as studied in this work, is there still a similar Lorentz invariance argument? If not, would it be possible to find some energetic evidence of confinement-deconfinement transition in this particular example? Maybe this is related to the subsequent relaxation time result.

I have also spotted a typo. When introducing the vectorized density matrix in the double Hilbert space (second paragraph of Section 3), there is an extra summation symbol $\sum_{mn}$.

Recommendation

Publish (easily meets expectations and criteria for this Journal; among top 50%)

---

## Round 1 · Referee Report · Anonymous (Referee 2) · 2024-11-5

Report

This study explores solvable models of steady-state topological orders, examining their stability under perturbations and analyzing both confinement transitions and relaxation dynamics. The results are substantial, comprehensive, and well-grounded, making a strong case for publication in SciPost.

I have a minor question regarding the characterization of topological order in mixed states through higher-form symmetry breaking, which may be either strong or weak depending on the conditions. Specifically, the authors observe that while steady-state topological degeneracy is susceptible to noise in two dimensions, it remains stable in three dimensions, where a true many-body phase with topological degeneracy emerges. Could a Mermin-Wagner-type argument, based on higher-form symmetry and dimensionality, provide insight into which dimensions and d-form symmetry breakings (for mixed-state topological order) yield stability?

Recommendation

Publish (meets expectations and criteria for this Journal)

---

## Editorial Decision

published